# The Preventive Mechanism of Anserine on Tert-Butyl Hydroperoxide-Induced Liver Injury in L-02 Cells via Regulating the Keap1-Nrf2 and JNK-Caspase-3 Signaling Pathways

**DOI:** 10.3390/md21090477

**Published:** 2023-08-29

**Authors:** Ming Chen, Jing Luo, Hongwu Ji, Wenkui Song, Di Zhang, Weiming Su, Shucheng Liu

**Affiliations:** 1College of Food Science and Technology, Guangdong Ocean University, Zhanjiang 524088, China; mingc09@126.com (M.C.); 18878477781@163.com (J.L.); 19867028119@163.com (W.S.); zjs578180838@sina.com (D.Z.); hdsuwm@163.com (W.S.); lsc771017@163.com (S.L.); 2Guangdong Provincial Key Laboratory of Aquatic Product Processing and Safety, Zhanjiang 524088, China; 3Guangdong Province Engineering Laboratory for Marine Biological Products, Zhanjiang 524088, China; 4Guangdong Provincial Engineering Technology Research Center of Marine Food, Zhanjiang 524088, China; 5Key Laboratory of Advanced Processing of Aquatic Product of Guangdong Higher Education Institution, Zhanjiang 524088, China; 6Collaborative Innovation Center of Seafood Deep Processing, Dalian Polytechnic University, Dalian 116034, China

**Keywords:** anrerine, liver injury, L-02 cell, oxidative stress, apoptosis, Keap1-Nrf2, JNK-Caspase-3

## Abstract

Anserine is a naturally occurring histidine dipeptide with significant antioxidant activities. This study aimed to investigate the preventive mechanism of anserine on tert-butyl hydroperoxide (TBHP)-induced liver damage in a normal human liver cell line (L-02 cells). The L-02 cells were pretreated with anserine (10, 20, and 40 mmol/L) and then induced with 400 μmol/L of TBHP for 4 h. The results showed that the survival rates of L-02 cells and the contents of GSH were significantly increased with the pretreatment of anserine; the activities of alanine aminotransferase (ALT) and aspartate aminotransferase (AST) in the extracellular fluid were sharply decreased; and the formation of reactive oxygen species (ROS), nuclear fragmentation, and apoptosis were significantly inhibited. In addition, anserine could bind to the Kelch domain of Kelch-like ECH-associated protein 1 (Keap1) with a binding force of −7.2 kcal/mol; the protein expressions of nuclear factor-erythroid 2-related factor-2 (Nrf2), quinone oxidoreductase 1 (NQO1), heme oxygenase-1 (HO-1), and Bcl-2 were upregulated by anserine in TBHP-induced L-02 cells, with the downregulation of p-JNK and caspase-3. In conclusion, anserine might alleviated liver injury in L-02 cells via regulating related proteins in the Keap1-Nrf2 and JNK-Caspase-3 signaling pathways.

## 1. Introduction

Oxidative stress refers to an imbalanced state between oxidative and antioxidant effects within the body’s the defense system [1]. Oxidative stress is associated with the pathophysiology and pathogenesis of many types of diseases, such as diabetes, atherosclerosis, hyperuricemia, and cardiovascular disease [2,3,4]. 

Nuclear factor erythroid 2-related factor 2 (Nrf2) is the main nuclear transcription factor that resists oxidative stress [5]. Under normal physiological conditions, Nrf2 is combined with Keap1, which is a negative regulator of Nrf2 that can induce the ubiquitination and degradation of Nrf2 [6]. Under oxidative stress, Nrf2 is dissociated from the complex and transferred into the nucleus to regulate the production of quinone oxidoreductase 1 (NQO1) and heme oxygenase-1 (HO-1) and to prevent oxidative damage [7]. Moreover, c-Jun N-terminal protein kinase (JNK) can be activated by a variety of stress factors, with the role of mediating apoptosis [8]. Under oxidative stress, JNK is phosphorylated and mediates downstream factor Bcl-2 expression, which then exerts proapoptotic effects through Caspase-3 (Cas-3) [9]. Thus, inhibiting the expression and activation of JNK protein in cells during stress can significantly reduce cell apoptosis, playing an important role in the treatment and prevention of diseases.

The liver, as an important metabolic organ, is the major target of oxidative stress [10]. Many studies have demonstrated that dysregulated oxidative stress inhibits the defense function of the liver, leading to massive liver cell damage, thereby inducing various liver diseases [11,12]. Thus, it is important to prevent or treat injured liver cells. At present, a number of drugs with anti-oxidative and anti-inflammatory properties have been evaluated as hepatoprotective agents. For example, the newly synthesized eicosapentaenoic acid esterifified 12-hydroxy stearic acid with a low cytotoxicity is resistant against oxidative stress in human hepatoma-derived cells by activating Nrf2 expression, thereby protecting hepatocytes [13,14]. Natural triterpenoids have exhibited promising hepatoprotective effects against H_2_O_2_-induced damage though their antioxidant properties on a primary cultured rat hepatocyte [15]. However, they generally exhibit a limited efficacy in the therapy of liver injury, and it is thus necessary to explore more potential agents for the treatment of liver injury [16].

Anserine (Figure 1) belongs to a family of histidyl dipeptides, consisting of β-alanine and 1-methyl-histidine. Anserine is widely distributed in the muscles and brains of vertebrates, especially in marine fish and birds. Anserine is a multifunctional dipeptide that exhibits the potential for anti-hyperuricemia, antioxidant, and anti-fatigue properties [17,18,19]. For example, anserine was found to have a protective effect under diabetic conditions, and short-term anserine treatment in diabetic mice improves glucose homeostasis and nephropathy [20]. Moreover, the improvement effect of anserine on mild cognitive impairment in healthy older adults was studied by Masuoka et al. [21]. In addition, our previous study found that anserine can not only reduce uric acid levels, but it can also effectively repair hyperuricemia-induced liver damage in rats [22]. However, the protective mechanism of anserine on the liver is not clear. To the best of our knowledge, the effect of anserine on liver injury in vitro has also not yet been studied.

The purposes of this study were to (1) establish a model of TBHP-induced liver injury in L-02 cells, (2) verify the preventive effect of anserine on liver injury in L-02 cells, and (3) clarify the mechanisms through which anserine alleviates liver injury in L-02 by regulating Keap1-Nrf2 and JNK-Caspase-3 signaling pathways. 

## 2. Results

### 2.1. Effects of TBHP on the Survival Rate of L-02 Cells

TBHP is a strong oxidant and is often used as an inducer for in vitro oxidative stress models [23]. The metabolism of TBHP will produce a large amount of ROS, which causes oxidative damage and even necrosis of cells [24]. In this study, TBHP was used to induce oxidative damage in L-02 cells to establish the injury model. The results are shown in Table 1; under the same TBHP treatment time, the survival rate of L-02 cells was gradually decreased with the increase in TBHP concentration. The cell survival rates of L-02 cells treated with TBHP for 4 h were approximately 72%, 58%, 44%, 39%, and 31%, with concentrations of 200, 400, 600, 800, and 1000 µmol/L, respectively. In this study, it was found that when the cell survival rate was between 50% and 60%, the model of TBHP-induced L-02 injury was more stable and more conducive for the next step of the experiment. According to the effect of different TBHP treatment times and different TBHP concentrations on the cell survival rate, the TBHP concentration of 400 µmol/L and TBHP treatment time of 4 h were selected as modeling conditions for the injured L-02 cells.

### 2.2. Protective Effect of Anserine on L-02 Cell Injury

When investigating the effects of drugs or active substances on cell growth based on certain functional characteristics, the toxic side effect and proliferation of the substance on cells are important factors affecting the experimental results. In this experiment, the toxicity and proliferation of anserine on L-02 cells were investigated by detecting the cell survival rate. As shown in Figure 2A, compared with the control group, the cell survival rates of L-02 cells showed no significant difference upon the addition of various concentrations of anserine (5–80 mmol/L) for 12 h. This result indicates that anserine had no toxic effect on L-02 cells in the concentration range of 5–80 mmol/L after 12 h. In addition, anserine, as an easily absorbed dipeptide, is a nutritional substrate for the growth of L-02; it did not promote the proliferation of L-02 cells in this range. 

Based on the above results, we next examined the effects of anserine pretreatment on TBHP-induced L-02 injury. As shown in Figure 2B, compared with the control group, the survival rate of L-02 cells in the model group was significantly decreased upon treatment with TBHP for 4 h. Compared with the model group, the survival rate of L-02 cells significantly increased after pretreatment with anserine in the concentration range of 5–80 mmol/L, and showed a concentration dependence in the concentration range of 5–40 mmol/L. The results indicate that anserine exhibited a protective effect on L-02 cells injured by TBHP.

### 2.3. Effect of Anserine on Biochemical Indicators in TBHP-Induced Damaged L-02 Cells

In this study, TBHP-induced damaged L-02 cells resulted in the release of a large amount of ALT and AST outside the cells, and the levels of ALT and AST in the cell culture medium were sharply increased. As shown in Figure 3A,B, compared with the control group, the enzyme activities of ALT and AST were increased by 113.8% and 64.3%, respectively. Upon pretreatment with different dosages of anserine and Vc, the increase in ALT and AST levels caused by TBHP were decreased to varying degrees. Compared with the model group, the enzyme activities of ALT in A-10, A-20, A-40, and Vc were decreased by 18.2%, 29.8%, 32.4%, and 42.7%, respectively, as well as displaying a 19.5%, 32.6%, 21.7%, and 28.2% decrease in AST, respectively. In addition, the protective effect of anserine on the nucleus of L-02 cells was also investigated. The L-02 cells were stained with DAPI and inspected using a fluorescence microscope. The normal cells had complete nuclei and uniform chromatin with a high fluorescence intensity. The chromatin of injured cells decreased to form many granular substances, and further nuclear fission could possible occur, which would have reduced the fluorescence intensity. The results show that TBHP treatment caused the decomposition of nuclear DNA and the pretreatment of anserine reversed this phenomenon (Figure 4). 

The liver injury in L-02 cells was mainly induced by the oxidative stress from TBHP. The content of ROS and the level of antioxidant GSH were tested to evaluate the antioxidative effect of anserine. As shown in Figure 3C, the level of ROS in the model group increased sharply after the treatment of TBHP. Compared with the model group, pretreatment with anserine and Vc significantly restrained the generation of ROS; the ROS levels in A-10, A-20, A-40, and Vc decreased by 32.8%, 18.8%, 29.2%, and 36%, respectively. In addition, the level of GSH was significantly increased with the pretreatment of anserine, which is one of the reasons for the decrease of ROS in L-02 cells. Compared with the model group, the contents of GSH in A-10, A-20, A-40, and Vc increased by 11.3%, 28.3%, 22.6%, and 29.4%, respectively (Figure 3D).

These above results from the various biochemical indicators indicate that anserine effectively protected against TBHP-induced L-02 cell injury. 

### 2.4. The Positive Regulation of Anserine on the Keap1-Nrf2 Signaling Pathway in L-02 Cells

To further investigate the mechanism of anserine protection against TBHP-induced oxidation emergency in L-02 cells, the effect of anserine on the Keap1-Nrf2 signaling pathway was studied. The molecular docking between anserine and Keap1 showed that anserine could bind to the Kelch domain of Keap1 with a binding force of −7.2 kcal/mol and form a hydrogen bond with VAL463, VAL512, ILE559, and VAL606 of Keap1, which indicated anserine may inhibit the ubiquitination of Nrf2 (Figure 4B and Figure 5A). In addition, the effect of anserine on the protein expression of Nrf2, NQO1, and HO-1 in the Keap1-Nrf2 signaling pathway were determined using Western blot. The results are shown in Figure 5C–F; compared with the model group, the protein expression levels of Nrf2, NQO1, and HO-1 were significantly increased to varying degrees in the anserine and Vc groups, and the promoting effect of anserine on the protein expression of Nrf2 and NQO1 was similar to that of Vc.

### 2.5. Inhibition of L-02 Cell Apoptosis by Anserine

The effect of anserine on the TBHP-induced apoptosis of L-02 cells was detected using flow cytometry. The results are shown in Figure 6; compared with the control group, the amount of apoptotic L-02 cells was sharply increased after treatment with TBHP, with no significant changes in dying cells. Compared with the model group, the contents of apoptotic L-02 cells in A-10, A-20, and A-40 decreased by 26.5%, 28.2%, and 42.9%, respectively. This result illustrates that anserine pretreatment inhibited the TBHP-induced unconventional apoptosis of L-02 cells.

### 2.6. Effect of Anserine on the JNK-Bcl-2-Caspase-3 Signaling Pathway in L-02 Cells

The JNK signaling pathway is an important branch of the MAPK pathway, which plays an important regulatory role in the process of cell apoptosis [25]. In this study, the effects of anserine on the protein expression of JNK, P-JNK, Bcl-2, Bax, and Caspase-3 were determined through Western blot. The results show that (Figure 7) compared with the control group, the protein expression of P-JNK was significantly increased in the model group. After pretreatment with anserine, the phosphorylation of JNK was significantly inhibited, leading to upregulation of the Bcl-2 protein expression and downregulation of Bax and Cas-3 protein expression, thereby inhibiting the apoptosis of L-02 cells. Compared with Vc, a high dose anserine exhibited a better regulatory ability. 

## 3. Discussion

Liver damage is caused by many different liver conditions and serves as a shared pathological foundation for various types of liver ailments [26,27]. In this study, the preventive effect of anserine on liver damage in L-02 cells and its reaction mechanisms were investigated. When L-02 cells were treated with 400 µmol/L of TBHP for 4 h, the L-02 cell injury model was successfully established. The results showed that anserine could effectively alleviate L-02 cell injury induced by TBHP, and no significant proliferative effects or toxic side effects were found for treatment with 5–80 mmol/L of anserine in L-02 cells within 12 h. Similarly, Cho BO, et al. successfully established a model of TBHP-induced liver injury in HepG2 cells to evaluate the liver injury repair function of kushenol C [28]. Rutin has also been proven to attenuate oxidative damage induced by TBHP in HepG2 cells [29].

Liver damage can increase the permeability of liver cell membranes, resulting in the release of ALT and AST into the extracellular space. Hence, ALT and AST are often used as key indicators to determine liver injury [30]. Accumulating evidence reveals that ROS-induced oxidative stress can cause various human illness, especially liver injury [31]. An effective method to resist liver injury is through the production of excessive ROS. In this study, after the induction of TBHP, the levels of ALT and AST in the cell culture medium were rapidly increased, resulting in significant DNA damage in the nucleus, indicating that the L-02 cells had been damaged. This damage may be caused by the production of a large amount of ROS from TBHP treatment. After pretreatment with anserine, the production of ROS was significantly reduced, the levels of ALT and AST were decreased, and the survival rate of L-02 cells was significantly improved, indicating that anserine could effectively alleviate TBHP-induced L-02 cell injury. Earlier research on hydrogen-peroxide-induced cell oxidative damage showed similar results [32]. 

Nrf2 plays a key role in modulating cellular defense against oxidative stress by regulating the antioxidant factors HO-1 and NQO1 [7]. Normally, Nrf2 is retained in the cytoplasm by binding to its negative regulator, Keap1. Keap1 mainly has two conservative domains, BTB and Kelch; the BTB site binds to ubiquitin ligase and the Kelch domain binds to Nrf2 [33]. Therefore, the Kelch domain is the main site for virtual screening of potential Nrf2 agonists. This study, through molecular docking, found that anserine could bind to the Kelch domain of Keap1 with a binding force of −7.2 kcal/mol and form hydrogen bonds with VAL463, VAL512, ILE559, and VAL606 of Keap1. In addition, anserine significantly increased the protein expression of Nrf2, NQO1, and HO-1 in L-02 cells. These results show that anserine might bind to the Kelch domain of Keap1 to prevent Nrf2 from ubiquitination and degradation. A previous study found that tangeretin maintained its antioxidant activity by reducing Nrf2 ubiquitination [34]. Under the stimulation of ROS generated by TBHP treatment, the accumulated Nrf2 in the cytoplasm of L-02 cells was transferred into the nucleus and upregulated the expression of downstream factors NQO1 and HO-1, thereby alleviating liver cell damage. 

Normal cell apoptosis is an autonomous and programmed cell death regulated by genes, but factors such as oxidative stress, non-polar heat stress, ultraviolet or ion radiation, and heavy metal stress can increase the rate of cell apoptosis [35,36]. Previous studies have proven that JNK is a stress-activated protein kinase that can be activated by various stress factors, with a mediating effect on cell apoptosis, and it is associated with the occurrence of various diseases [37]. In this study, the apoptosis of L-02 cells was significantly increased by TBHP treatment through the detection of flow cytometry. The protein expressions of p-JNK and Cas-3 were significantly increased, while the expression of the Bcl-2 protein was significantly reduced. After pretreatment with anserine, these changes were significantly reversed. In addition, the content of GSH in L-02 cells obviously increased. These results show that anserine may restrain the phosphorylation of JNK, thereby reducing the phosphorylation of Bcl-2 by JNK and inhibiting GSH leakage. All in all, anserine may alleviate liver injury by inhibiting cell apoptosis. Interestingly, oleanolic acid displayed similar effects in QZG cells via an increase in the generation of antioxidants and by regulating the expression of p-JNK [38].

## 4. Materials and Methods

### 4.1. Chemicals and Reagents

Anserine (99%) was obtained from Beijing Solarbio Science and Technology Co., Ltd. (Beijing, China). Vitamin C (Vc) and TBHP (9 mmol/mL) were obtained from Sigma Aldrich (Shanghai, China). L-02 (HL7702) cells were purchased from Wuhan Shangen Biotechnology Co., Ltd. (Wuhan, China). Dulbecco’s Modified Eagle Medium (DMEM), phosphate buffered saline (PBS), fetal bovine serum (FBS), PBS buffer, penicillin, streptomycin, and trypsin were all purchased from Gibco (Suzhou, China). Assay kits for determining ALT, AST, ROS, and GSH were purchased from the Jiancheng Institute of Biotechnology (Nanjing, Jiangsu, China). RIPA Lysis Buffer and Phenylmethanesulfonyl fluoride (PMSF) and the Cell Counting Kit-8 (CCK-8) were purchased from Beyotime Biotechnology Co., Ltd. (Shanghai, China). The Annexin V-FITC Apoptosis Detection Kit was purchased from Solarbio Biotechnology Co., Ltd. (Beijing, China). β-Actin (sc-47778, mouse monoclonal antibody), Nrf2 (sc-365949, mouse monoclonal antibody), NQO1 (sc-32793, mouse monoclonal antibody), JNK (sc-7345, mouse monoclonal antibody), p-JNK (sc-6254, mouse monoclonal antibody), Bcl-2 (sc-7382, mouse monoclonal antibody), Bax (sc-7480, mouse monoclonal antibody), and Caspase-3 (sc-7272, mouse monoclonal antibody) were provided by Santa Cruz Biotechnology Company (Santa Cruz, CA, USA).

### 4.2. Cell Culture

L-02 cells (Tongpai Biotechnology Co., Ltd., Shanghai, China) were cultured in DMEM supplemented with 12% (*v/v*) FBS, 100 U/mL penicillin, and 100 µg/mL streptomycin at 37 °C in a 5% CO_2_ atmosphere. After PBS washing and trypsin digestion, the L-02 cells were centrifuged at 1000 rpm for 5 min (Eppendorf 5810R, Eppendorf, Germany), then resuspended and passaged at a ratio of 1:4.

### 4.3. Establishment of TBHP Induced L-02 Cell Injury Model

L-02 cells in a logarithmic growth period were seeded at a density of 2 × 10^4^ cells per well and cultured in 96-well plates for 24 h. TBHP was proportionally diluted with DMEM supplemented with 12% (*v/v*) FBS. Then, the L-02 cells were treated with 200, 400, 600, 800, and 1000 μmol/L of TBHP, and the cell viability was measured by CCK-8 at 2 h, 4 h, 8 h, 12 h, and 24 h, respectively.

### 4.4. Cytotoxicity and Proliferation Assay

L-02 cells in the logarithmic growth period were seeded at a density of 2 × 10^4^ cells per well and cultured in 96-well plates for 12 h. Then, the L-02 cells were treated with 5, 10, 20, 40, and 80 mmol/L of anserine for 12 h, and the cell viability was measured by CCK-8 at an absorbance of 450 nm using a microplate reader (Varioskan Flash, Thermo, Waltham, MA, USA). The groups A-5, A-10, A-20, A-40, and A-80 represent the anserine concentrations of 5 mmol/L, 10 mmol/L, 20 mmol/L, 40 mmol/L, and 80 mmol/L, respectively.

### 4.5. Cytoprotective Effect of Anserine

L-02 cells in the logarithmic growth period were seeded at a density of 2 × 10^4^ cells per well and cultured in 96-well plates for 12 h. Then, the L-02 cells were treated with 5, 10, 20, 40, and 80 mmol/L of anserine for 12 h and then treated with 400 μmol/L of TBHP for 4 h. The cell viability was measured using CCK-8.

### 4.6. Measurement of ALT, AST, ROS, and GSH in TBHP-Induced L-02 Cells

L-02 cells in the logarithmic growth period were seeded at a density of 4 × 10^5^ cells per well and cultured in 6-well plates for 12 h. Then, the L-02 cells were treated with anserine (10, 20, and 40 mmol/L) and Vc (50 mg/L) for 12 h and then treated with 400 μmol/L of TBHP for 4 h. The supernatant was used for detecting the enzymatic activity of ALT and AST using their respective activity assay kits. The L-02 cell were lysed using a lysis buffer and its contents were collected for the evaluation of ROS and GSH using their respective activity assay kits. The groups A-10, A-20, and A-40 represent the anserine concentrations of 10 mmol/L, 20 mmol/L, and 40 mmol/L, respectively.

### 4.7. Fluorescence Observation of L-02 Nucleus Using DAPI Stain

L-02 cells in the logarithmic growth period were seeded at a density of 4 × 10^5^ cells per well and cultured in 6-well plates for 12 h. Then, the L-02 cells were treated with anserine (10, 20, and 40 mmol/L) and Vc (50 mg/L) for 12 h and then treated with 400 μmol/L of TBHP for 4 h. The cell culture medium was removed and an appropriate amount of DAPI staining was added for 5 min, then washed with PBS three times, and finally observed under an inverted microscope (DMI4000B, Leica, Wetzlar, Germany).

### 4.8. Molecular Docking

The Keap1 3D structure (Protein Data Bank (PDB) ID: 6QMC) was obtained from PDB and the water and ligands were removed from the original spatial structure to prepare a molecular docking simulation protein for Nrf2. The anserine 3D structure (PubChem CID: 112072) was obtained from PubChem. The Autodock Vina was used for the docking of Keap1 and anserine and Vision 1.5 was used for analyzing their binding interactions.

### 4.9. Determination of Apoptosis

L-02 cells in the logarithmic growth period were seeded at a density of 4 × 10^5^ cells per well and cultured in 6-well plates for 12 h. Then, the L-02 cells were treated with anserine (10, 20, and 40 mmol/L) and Vc (50 mg/L) for 12 h and then treated with 400 μmol/L of TBHP for 4 h. The L-02 cells were collected and added in FITC-Annexin V and propidium iodide for determining in the flow cytometry (FACS Aria, Becton, Dickinson and Company, Franklin Lakes, NJ, USA).

### 4.10. Western Blot

L-02 cell samples in the same protein concentration were mixed with 5× loading buffers and treated in a boiling water bath for 5 min, and separated by SDS-PAGE protein electrophoresis, then transferred to a 0.22 μm PVDF membrane. After blocking for 10–15 min in a quick Western blocking solution, the membranes were incubated for 12 h at 4 °C with the primary antibodies (1:1000). Then, the membranes were incubated with a secondary antibody (1:2000) for 2 h. Finally, immunoreactive bands were observed by chemiluminescence and quantified by Image J.

### 4.11. Statistical Analysis

The data were shown as mean ± SD with statistical analysis performed used JMP14.0 software (SAS Institute, Cary, NC, USA) and graphic rendering using Origin 9.0 software (OriginLab, Northampton, MA, USA). All of the experiments were carried out at least in triplicate.

## 5. Conclusions

In summary, we successfully demonstrated the potent protective effect of anserine on TBHP-induced L-02 cell injury. The mechanism underlying this effect was realized by regulating the Keap1-Nrf2 and JNK-Caspase-3 signaling pathways. These findings suggest that anserine may be used as a potential protectant of the prevention and treatment of liver injury.

## Figures and Tables

**Figure 1 marinedrugs-21-00477-f001:**
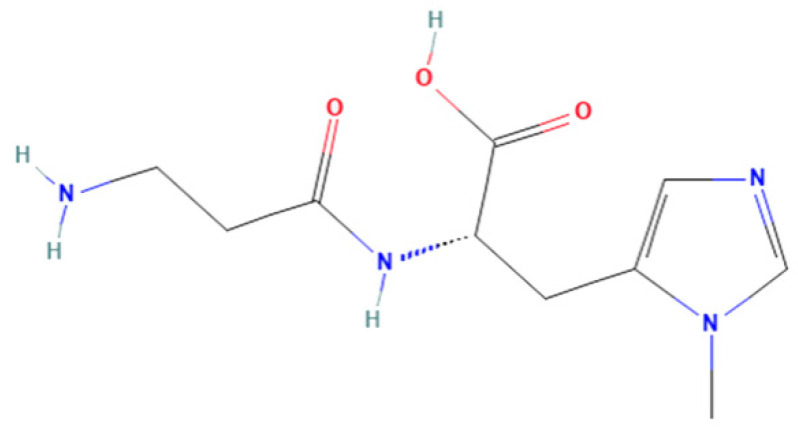
Chemical structures of anserine.

**Figure 2 marinedrugs-21-00477-f002:**
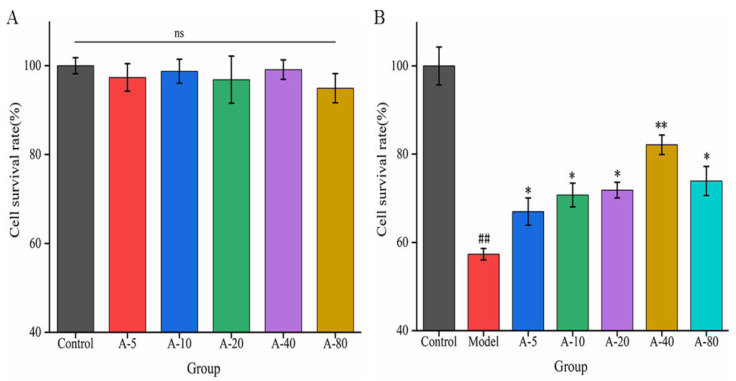
The effect of anserine on the survival rate of L-02 cells (**A**) and the survival rate of TBHP-induced L-02 cells (**B**). Data are displayed as the mean ± SD (*n* = 3). ^##^
*p* < 0.01 compared with the control group; * *p* < 0.05, ** *p* < 0.01 compared with the model group.

**Figure 3 marinedrugs-21-00477-f003:**
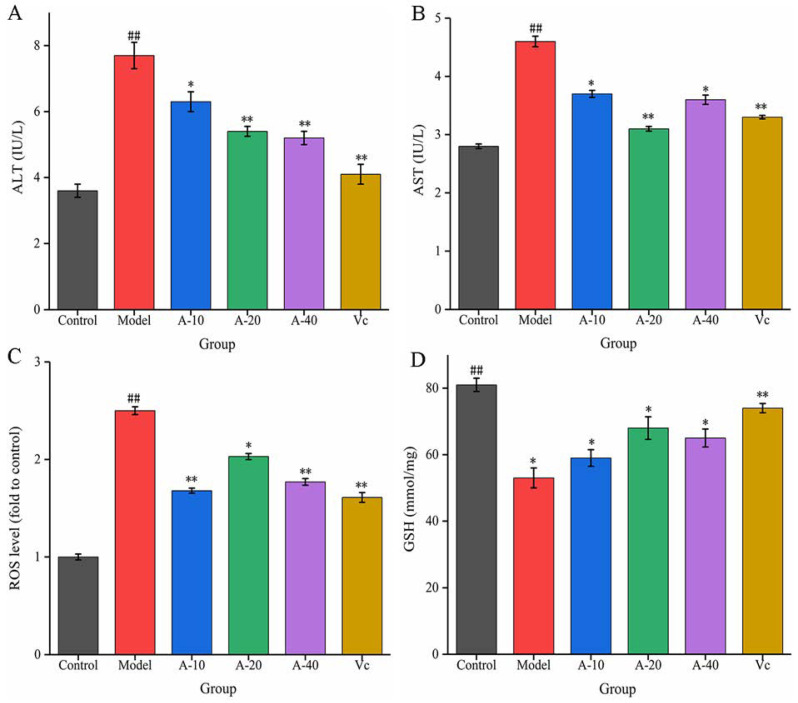
Effect of anserine on ALT, AST, ROS, GSH, and the nucleus. (**A**) ALT, (**B**) AST, (**C**) ROS, and (**D**) GSH. Data are displayed as mean ± SD (*n* = 3). ^##^ *p* < 0.01 compared with the control group; * *p* < 0.05, ** *p* < 0.01 compared with the model group.

**Figure 4 marinedrugs-21-00477-f004:**
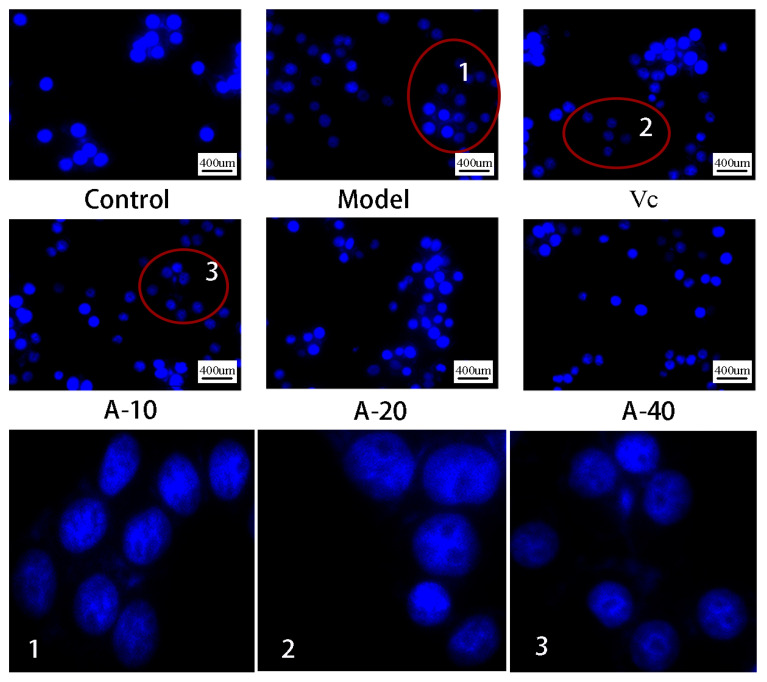
DAPI staining and fluorescence observation of L-02 cells.

**Figure 5 marinedrugs-21-00477-f005:**
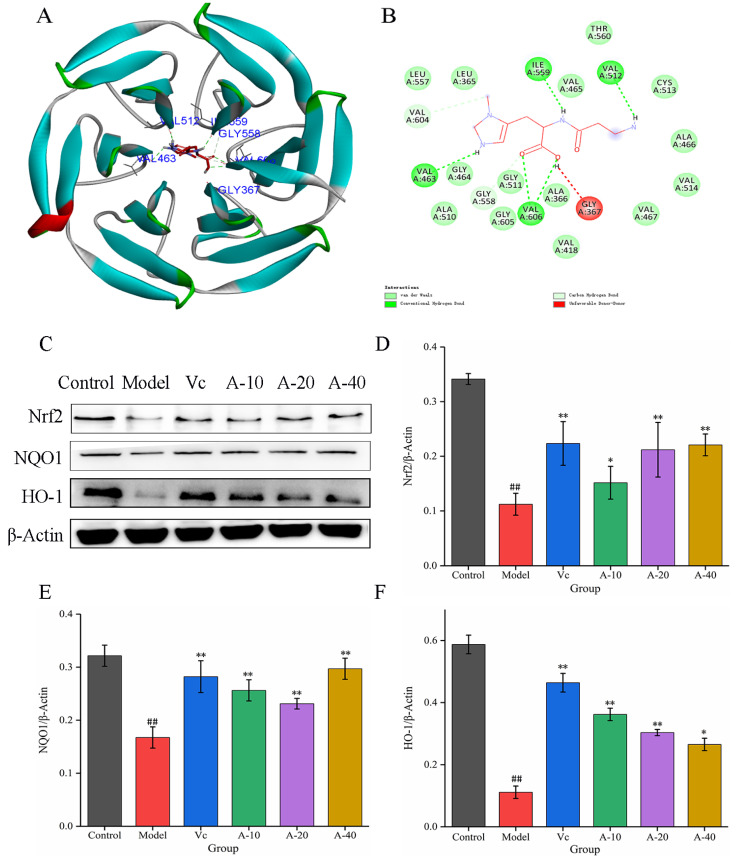
Molecular docking between anserine and Keap1 and the expressions of Nrf2, HO-1, and NQO1 in L-02 cells. (**A**) The proposed docking mode of anserine binding with the Keap1 Kelch domain. (**B**) The binding modes of anserine with Keap1 Kelch domain showing interacting amino acids and H-bonds. (**C**–**F**) The expressions of Nrf2, HO-1, and NQO1 in L-02 cells. Data are displayed as mean ± SD (*n* = 6). ^##^
*p* < 0.01 compared with the control group; * *p* < 0.05, ** *p* < 0.01 compared with the model group.

**Figure 6 marinedrugs-21-00477-f006:**
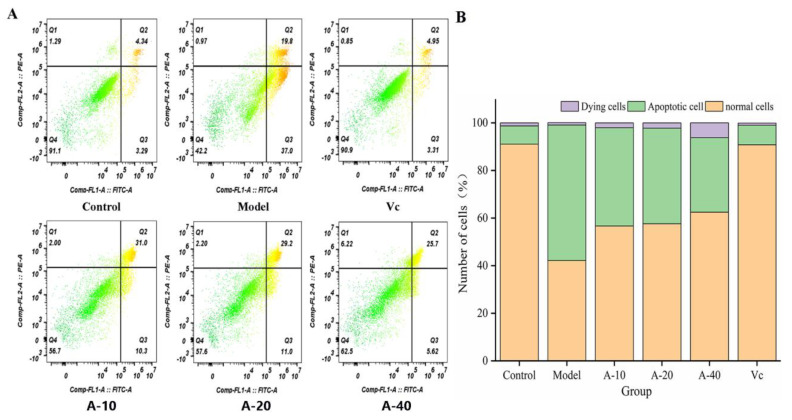
Delaying effect of anserine on the apoptosis of L-02 Cells. (**A**) Observations using a flow cytometer. (**B**) Number of apoptotic cells.

**Figure 7 marinedrugs-21-00477-f007:**
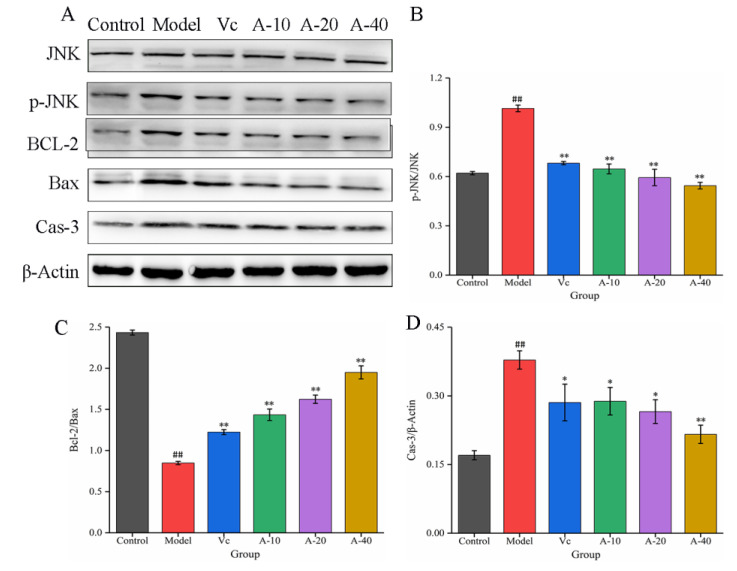
The expressions of JNK, p-JNK, BCL-2, Bax, and Cas-3 in L-02 cells. (**A**) Western blot bands of JNK, p-JNK, BCL-2, Bax, and Cas-3. (**B**–**D**) Quantitative analysis of p-JNK/JNK, Bcl-2/Bax, and Cas-3. Data are displayed as mean ± SD (*n* = 3). ^##^
*p* < 0.01 compared with the control group; * *p* < 0.05, ** *p* < 0.01 compared with the model group.

**Table 1 marinedrugs-21-00477-t001:** The effect of TBHP concentration and modeling time on the survival rate of L-02 cells. Data are displayed as mean ± SD (*n* = 3).

Concentration (μmol/L)	Cell Survival Rate (%)				
	2 h	4 h	8 h	12 h	24 h
0	100 ± 4.17	100 ± 1.58	100 ± 1.06	100 ± 1.46	100 ± 1.21
200	74.04 ± 3.91	71.82 ± 1.14	66.77 ± 1.21	49.35 ± 1.24	30.04 ± 0.81
400	65.79 ± 5.43	57.85 ± 0.71	44.13 ± 1.13	40.76 ± 1.28	27.16 ± 0.95
600	49.02 ± 3.76	43.86 ± 1.11	36.91 ± 1.12	37.62 ± 1.25	23.61 ± 0.63
800	44.86 ± 4.12	38.61 ± 0.81	35.44 ± 1.01	31.79 ± 1.28	21.03 ± 0.25
1000	39.36 ± 2.92	30.81 ± 0.73	28.68 ± 1.24	25.09 ± 0.76	21.48 ± 0.67

## Data Availability

Data available upon request from the authors.

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
