# Peer review of "The Preventive Mechanism of Anserine on Tert-Butyl Hydroperoxide-Induced Liver Injury in L-02 Cells via Regulating the Keap1-Nrf2 and JNK-Caspase-3 Signaling Pathways"

_marinedrugs, 2023, doi:10.3390/md21090477_

Round 1
Reviewer 1 Report (Previous Reviewer 1)
The author made improvements for my comments. I accept this manuscript. However, I instruct you to make detailed corrections.
Line82 Effects of TPHP → TBHP
Line89 Author indicated that 71%, 54%, 47%, 38%, and 30%. However, I checked the Table1. The numbers are different, but which is correct?
71% → about 72% (71.82%)
54% → about 58% (57.85%)
47% → about 44% (43.86%)
38% → about 38% (38.61%)
30% → about 31% (30.81%):
Minor editing of English language required
Author Response
We would like to express our sincere thanks to you for the constructive and positive comments. The errors you pointed out have been corrected in the manuscript. The cell survival rates of L-02 cells treated with TBHP for 4 h were approximately 72%, 58%, 44%, 39%, and 31%, with the concentrations of 200, 400, 600, 800, 1000 µmol/L, respectively.
Reviewer 2 Report (Previous Reviewer 2)
The authors have considered my comments and significantly revised their manuscript compared to the previous version. Since the study is interesting as the authors aimed to investigate the protective effects of anserine using a liver injury in vitro model. I recommend the work for the publication in the journal.
Not applicable
Author Response
We would like to express our sincere thanks to your constructive and positive comments to improve the manuscript quality.
Reviewer 3 Report (New Reviewer)
The manuscript "The preventive mechanism of anserine on tert-butyl hydroperoxide-induced liver injury in L-02 cells via regulating the Keap1-Nrf2 and JNK-Caspase-3 signaling pathways", reports important effects of anserine. However, the authors should clarify the following points:
In all biological assays, positive controls (drugs) must be added to be able to compare our results obtained.
What were the criteria used to use the concentrations of 10, 20, and 40 mmol/L of Anserine?
The authors must review the entire manuscript, since the new version presents fragments written in British English and fragments in American English. They should make the entire manuscript uniform.
Author Response
We would like to express our sincere thanks to you for the constructive and positive comments. According to your comments, I made corresponding revisions in the manuscript (red font), and my responses to reviewer are as follow (blue font):
- In all biological assays, positive controls (drugs) must be added to be able to compare our results obtained.
Answer:
The vitamin C (Vc) was selected as a positive control substance in this study, and we added some description of the effect of Vc in the manuscript.
- What were the criteria used to use the concentrations of 10, 20, and 40 mmol/L of Anserine?
Answer:
In section 2.2, we examined the effects of 5-80 mmol/L of anserine on TBHP-induced L-02 injury. In the concentration range of 5-40 mmol/L, the higher the concentration of anserine, the higher the cell survival rate. In the preliminary experiment, we also set a concentration of 60 mmol/L. In the concentration range of 40-80 mmol/L, the higher the concentration of anserine, the lower the cell survival rate. Although there was no significant cytotoxicity of 5-80 mmol/L anserine to L-02 cells, high concentrations of carnosine may have an adverse effect on L-02 cells in TBHP-induced cell model, so we chose the concentration range of 0-40 mmol/L, and the concentrations of 10, 20, and 40 mmol/L of anserine had a better repair effect in this concentration range.

This manuscript is a resubmission of an earlier submission. The following is a list of the peer review reports and author responses from that submission.
Round 1
Reviewer 1 Report
This manuscript has a serious point. Therefore, I cannot peer review. The details are as follows.
Comment 1
In Figure 2, author used concentration of anserine (50-80 mg/ml). However, in figure legends, author indicated 5~80 mmol/mL. Which is correct? And what is Vc? No description for Vc.
Comment 2
Author indicated the inhibiting ubiquitination of Nrf2. I can’t see any results to show that. Author checked the increasing of t-BHP induced ubiquitinated proteins and decreasing the ubiquitination of Nrf2. And also, Nrf2 activity is determined by nuclear translocation. Because I can’t see that Nrf2, HO-1, NQO1 expressions is not altered by t-BHP treatment compared to CTL. If there is a difference as well as a graph (Fig. 4D-F), please match the picture (Fig. 4C) with the graph (Fig. 4D-F). This is very important point.
Comment 3
Author measured p-JNK, Bcl2, Cas-3 in Figure 6. This is not enough. Author should be measure total JNK, Bax, Pro-caspase, cleaved caspase etc. Specially, author checked the total JNK.
Comment 4
Author should write in detail about the antibodies. And, I can’t see the results of keap1 expression. What is Keap1 that the author indicated in Chemicals and regents section?
No problem
Reviewer 2 Report
The article entitled “The preventive mechanism of anserine on tert-butyl hydroperoxide-induced liver injury in L-02 cells via inhibiting ubiquitination of Nrf2 and JNK phosphorylation” by Chen M et al deals with the investigation of effect of anserine on tert-butyl hydroperoxide (TBHP) induced liver injury in normal liver cells. The authors found that pretreatment of liver cells with anserine significantly increased the antioxidant related genes and protected the liver cells from TBHP induced damages. Although the study is interesting there are some corrections required.
Introduction: Well written. However, the previous studies on investigation of antioxidant mechanisms in liver cells should be discussed in brief (PMID 34299218, PMID 32397146) and also the term in vitro should be represented in italic throughout the manuscript.
-Remove the background of Anserine structure
Results:
Section 2.1: mention in what solvent the TBHP was prepared and stored
Table 1: the numbers are overlapped and difficult to understand. Is the data is mean+/SD. Indicate it appropriately.
Regarding anerine concentrations in the abstract, authors mention as mmol/L whereas in section 2.2 concentration in mg/mL. I recommend unifying the units for better understanding. It seems surprising that Anserine is even not toxic to cells at 80 mg/mL. in what solvent the compound is dissolved and how many numbers of cells were used? If it is 96 well then, the 80mg is quite large amount may interfere with the assay. If there any previous references showing less toxicity of Anserine recommend citing it.
-Could the authors elaborate the results of fluorescence assay. What does the red marks indicate? How authors confirmed the decomposition of nuclear DNA. The scale bar is missing in Figure 3E.
Figure 5 is unclear. The results of flow cytometry is essential. Could authors indicate the cell types with gates?
Figure 6: The data is n=3 is too low in number as the protein levels may vary
Section 4.1 : the primers of antioxidant genes used for the assays should be given.
4.2: Describe the source or supplier of L-02 cells.
4.3: cells numbers should be written in exponential forms: 2x104
4.4 again it is confusing that whether it is mmol/L or mg/mL
Minor spell checks need to be corrected.
Round 2
Reviewer 1 Report
There is no improvement yet. Therefore, I request a major revision again.
Comment 1
In Figure 2, author used concentration of anserine (50-80 mg/ml). However, in figure legends, author indicated 5~80 mmol/mL. Which is correct? And what is Vc? No description for Vc.
Answer 1: I apologize for this serious mistake. The concentration of anserine has been corrected to 5~80 mmol/L. And Vc is the abbreviation for vitamin C, which has been re-described in the chemicals and regents section.
→ I checked. It’s OK.
Comment 2 Author indicated the inhibiting ubiquitination of Nrf2. I can’t see any results to show that. Author checked the increasing of t-BHP induced ubiquitinated proteins and decreasing the ubiquitination of Nrf2. And also, Nrf2 activity is determined by nuclear translocation. Because I can’t see that Nrf2, HO-1, NQO1 expressions is not altered by t-BHP treatment compared to CTL. If there is a difference as well as a graph (Fig. 4D-F), please match the picture (Fig. 4C) with the graph (Fig. 4D-F). This is very important point.
Answer 2: 1) Undoubtedly, you are an expert in this field. To be honest, this was just a deduction, and we conducted some relevant experiments and listed some previous studies in the discussion to support our deduction. Under normal physiological conditions, Nrf2 is combined with Keap1 which is a negative regulator of Nrf2 that can induce the ubiquitination and degradation of Nrf2. The results of molecular docking in this study indicated that anserine could bind to the Kelch domain of Kelch-like ECH-associated protein 1 (Keap1), which may reduce the binding between Keap and Nrf2, thereby inhibiting the ubiquitination of Nrf2 and promoting the nuclear translocation of Nrf2.
→ The title is The preventive mechanism of anserine on tert-butyl hydroperoxide-induced liver injury in L-02 cells via inhibiting ubiquitination of Nrf2 and JNK phosphorylation. It’s easy. The authors have to measure the ubiquitination of Nrf2. You have to do the immunoprecipitation using anti-Nrf2 or ubiquitin antibodies. Why don't you do it? Is there a reason? If you don't measure this, you should change the title.
2) In a single target protein band, compared with the control group, the expression of Nrf2, HO-1, and NQO1 in the model group was not significant, because the protein concentration of samples varied in SDS-PAGE gel electrophoresis. Unfortunately, the protein concentration of the samples was difficult to be exactly the same, although we tried hard to adjust it. So we detected the expression of the internal reference protein β-actin in the sample to perform relative quantification of the target protein. In the Fig. 4C, the expression of β-actin was significantly different between the control group and model group. The Fig. 4D-F were depicted based on the ratio of target protein (Nrf2, HO-1, NQO1) to β-actin.
→ I checked all data (n1~3). Because I can’t see that Nrf2, HO-1, NQO1 expressions is not altered by t-BHP treatment compared to CTL. Please, show a typical band image and match the level of the bar below.
Comment 3 Author measured p-JNK, Bcl2, Cas-3 in Figure 6. This is not enough. Author should be measure total JNK, Bax, Pro-caspase, cleaved caspase etc. Specially, author checked the total JNK.
Answer 3: We have added the western blot bands of JNK and Bax according to your requirements and recalculated the ratio of p-JNK/JNK and Bcl2/Bax in the Figure 6.
→ I checked this. Please, show a typical band image and match the level of the bar below for Bax/Bcl2. I can't see that I can see that Bax expression is increased and Bcl2 isn’t affect in model cells. Additionally, in Fig. 4, author indicated “Unfortunately, the protein concentration of the samples was difficult to be exactly the same, although we tried hard to adjust it.” However, I can see equal protein concentration. Why didn't author perform relative quantification of the target protein using actin in Fig.5. Author measure another internal control (GAPDH, HPRT etc.)
Comment 4 Author should write in detail about the antibodies. And, I can’t see the results of keap1 expression. What is Keap1 that the author indicated in Chemicals and regents section? Answer 4: As per your request, we have added detailed information about the antibodies in chemicals and regents section. The results of keap1 expression was not shown in this study, we have already deleted it in chemicals and regents section. Thank you for your rigorous and patient review.
→ I checked this.
Minor comment
Figure 3. Effect of anserine on ALP, ALT, ROS, GSH and nucleus. (A): ALT, (B): AST, (C): ROS, (D): 172 GSH, (E): DAPI staining and fluorescence observation. Data are displayed as the mean ± SD (n =3).
→ What is ALP ?
There is no improvement yet. Therefore, I request a major revision again.
Comment 1
In Figure 2, author used concentration of anserine (50-80 mg/ml). However, in figure legends, author indicated 5~80 mmol/mL. Which is correct? And what is Vc? No description for Vc.
Answer 1: I apologize for this serious mistake. The concentration of anserine has been corrected to 5~80 mmol/L. And Vc is the abbreviation for vitamin C, which has been re-described in the chemicals and regents section.
→ I checked. It’s OK.
Comment 2 Author indicated the inhibiting ubiquitination of Nrf2. I can’t see any results to show that. Author checked the increasing of t-BHP induced ubiquitinated proteins and decreasing the ubiquitination of Nrf2. And also, Nrf2 activity is determined by nuclear translocation. Because I can’t see that Nrf2, HO-1, NQO1 expressions is not altered by t-BHP treatment compared to CTL. If there is a difference as well as a graph (Fig. 4D-F), please match the picture (Fig. 4C) with the graph (Fig. 4D-F). This is very important point.
Answer 2: 1) Undoubtedly, you are an expert in this field. To be honest, this was just a deduction, and we conducted some relevant experiments and listed some previous studies in the discussion to support our deduction. Under normal physiological conditions, Nrf2 is combined with Keap1 which is a negative regulator of Nrf2 that can induce the ubiquitination and degradation of Nrf2. The results of molecular docking in this study indicated that anserine could bind to the Kelch domain of Kelch-like ECH-associated protein 1 (Keap1), which may reduce the binding between Keap and Nrf2, thereby inhibiting the ubiquitination of Nrf2 and promoting the nuclear translocation of Nrf2.
→ The title is The preventive mechanism of anserine on tert-butyl hydroperoxide-induced liver injury in L-02 cells via inhibiting ubiquitination of Nrf2 and JNK phosphorylation. It’s easy. The authors have to measure the ubiquitination of Nrf2. You have to do the immunoprecipitation using anti-Nrf2 or ubiquitin antibodies. Why don't you do it? Is there a reason? If you don't measure this, you should change the title.
2) In a single target protein band, compared with the control group, the expression of Nrf2, HO-1, and NQO1 in the model group was not significant, because the protein concentration of samples varied in SDS-PAGE gel electrophoresis. Unfortunately, the protein concentration of the samples was difficult to be exactly the same, although we tried hard to adjust it. So we detected the expression of the internal reference protein β-actin in the sample to perform relative quantification of the target protein. In the Fig. 4C, the expression of β-actin was significantly different between the control group and model group. The Fig. 4D-F were depicted based on the ratio of target protein (Nrf2, HO-1, NQO1) to β-actin.
→ I checked all data (n1~3). Because I can’t see that Nrf2, HO-1, NQO1 expressions is not altered by t-BHP treatment compared to CTL. Please, show a typical band image and match the level of the bar below.
Comment 3 Author measured p-JNK, Bcl2, Cas-3 in Figure 6. This is not enough. Author should be measure total JNK, Bax, Pro-caspase, cleaved caspase etc. Specially, author checked the total JNK.
Answer 3: We have added the western blot bands of JNK and Bax according to your requirements and recalculated the ratio of p-JNK/JNK and Bcl2/Bax in the Figure 6.
→ I checked this. Please, show a typical band image and match the level of the bar below for Bax/Bcl2. I can't see that I can see that Bax expression is increased and Bcl2 isn’t affect in model cells. Additionally, in Fig. 4, author indicated “Unfortunately, the protein concentration of the samples was difficult to be exactly the same, although we tried hard to adjust it.” However, I can see equal protein concentration. Why didn't author perform relative quantification of the target protein using actin in Fig.5. Author measure another internal control (GAPDH, HPRT etc.)
Comment 4 Author should write in detail about the antibodies. And, I can’t see the results of keap1 expression. What is Keap1 that the author indicated in Chemicals and regents section? Answer 4: As per your request, we have added detailed information about the antibodies in chemicals and regents section. The results of keap1 expression was not shown in this study, we have already deleted it in chemicals and regents section. Thank you for your rigorous and patient review.
→ I checked this.
Minor comment
Figure 3. Effect of anserine on ALP, ALT, ROS, GSH and nucleus. (A): ALT, (B): AST, (C): ROS, (D): 172 GSH, (E): DAPI staining and fluorescence observation. Data are displayed as the mean ± SD (n =3).
→ What is ALP ?
Nothing